

1            Widespread and Persistent Ozone Pollution in Eastern China

Guohui Li[1*], Naifang Bei[2], Junji Cao[1*], Jiarui Wu[1], Xin Long[1], Tian Feng[1, 2], Wenting Dai[1], Suixin Liu[1], Qiang
Zhang[3], and Xuexi Tie[1]
[1]Key Lab of Aerosol Chemistry and Physics, SKLLQG, Institute of Earth Environment, Chinese Academy of
Sciences, Xi'an, China
[2]School of Human Settlements and Civil Engineering, Xi'an Jiaotong University, Xi'an, Shaanxi, China
[3]Department of Environmental Sciences and Engineering, Tsinghua University, Beijing, China
[*]Correspondence to: Guohui Li (ligh@ieecas.cn) and Junji Cao (jjcao@ieecas.cn)
**Abstract**: Rapid growth of industrialization, transportation, and urbanization has caused
increasing emissions of ozone ($O_3$) precursors recently, enhancing the $O_3$ formation in
Eastern China. We show here that Eastern China has experienced widespread and persistent
$O_3$ pollution from April to September in 2015 based on the $O_3$ observations in 223 cities. The
observed maximum 1-h $O_3$ concentrations exceed 200 μg m$^{-3}$ in almost all the cities, 400 μg
m$^{-3}$ in more than 25% of the cities, and even 800 μg m$^{-3}$ in six cities in Eastern China. The
average daily maximum 1-h $O_3$ concentrations are more than 160 μg m$^{-3}$ in 45% of the cities,
and the 1-h $O_3$ concentrations of 200 μg m$^{-3}$ have been exceeded on over 10% of days from
April to September in 129 cities. A widespread and severe $O_3$ pollution episode from 22 to 28
May 2015 in Eastern China has been simulated using the WRF-CHEM model to evaluate the
$O_3$ contribution of biogenic and various anthropogenic sources. The model generally
performs reasonably well in simulating the temporal variations and spatial distributions of
near-surface $O_3$ concentrations. Using the factor separate approach, sensitivity studies have
indicated that the industry source plays the most important role in the $O_3$ formation, and
constitutes the culprit of the severe $O_3$ pollution in Eastern China. The transportation source
contributes considerably to the $O_3$ formation, and the $O_3$ contribution of the residential source
is not significant generally. The biogenic source provides a background $O_3$ source, and also
plays an important role in the south of Eastern China. Further model studies are needed to
comprehensively investigate $O_3$ formation for supporting the design and implementation of
$O_3$ control strategies, considering rapid changes of emissions inventories and photolysis
caused by the 'Atmospheric Pollution Prevention and Control Action Plan', released by the
Chinese State Council in 2013.





## 1    Introduction


In the urban planetary boundary layer (PBL), ozone ($O_3$) is formed as a result of
photochemical reactions involving volatile organic compounds (VOCs) and nitrogen oxide
($NO_x$) in the presence of sunlight (Brasseur et al., 1999):
$$NO_2 + h\upsilon \rightarrow NO + O(^3P) \quad (290\ nm\ < \lambda < 420\ nm)$$
$$O(^3P) + O_2 + M \rightarrow O_3 + M$$
$$O_3 + h\upsilon \rightarrow O_2 + O(^1D) \quad (290\ nm\ < \lambda < 329nm)$$
$$O(^1D) + H_2O \rightarrow 2OH$$
$$OH + VOCs + O_2 \rightarrow RO_2 + others$$
$$RO_2 + NO \rightarrow RO + NO_2$$
where $h\upsilon$ represents the energy of a photo; $O(^3P)$ and $O(^1D)$ represent the ground state and
electronically excited oxygen atoms, respectively; $RO_2$, $RO$, and OH denote peroxy, oxy, and
hydroxyl radicals, respectively. High $O_3$ concentrations ($[O_3]$) are of major environmental
concerns due to its deleterious impacts on ecosystems (e.g., National Research Council, 1991)
and human health  (Lippman, 1993; Weinhold, 2008).
The emissions of $O_3$ precursors, VOCs and $NO_x$, have been significantly increased
recently in China due to rapid industrialization and urbanization, and increasing
transportation activity (e.g., Zhang et al., 2009; Kurokawa et al., 2013; Yang et al., 2015).
Satellite measurements have demonstrated that $NO_x$ emissions have been increased by a
factor of 2 in Central and East China from 2000 to 2006 (Richter et al., 2005). Zhang et al.
(2009) have also shown an increasing trend of $NO_x$ emissions with an enhancement of 55%
in China from 2001 to 2006. $NO_x$ emissions have still continued to increase since 2006,
caused by increasing power plants and vehicles (Wang et al., 2012; Wang et al., 2013; Yang
et al., 2015). VOCs emissions have been estimated to increase by 29% during 2001 – 2006 in





China (Zhang et al., 2009), and predicted to increase by 49% by 2020 relative to 2005 levels
(Xing et al., 2011).

Increasing $O_3$ precursors emissions has caused $O_3$ to be one of the most serious air

pollutants of concern during summertime, particularly in Eastern China, including the North
China Plain (NCP), Yangtze River Delta (YRD), and Pearl River Delta (PRD) (e.g., Xu et al.,
2011; Tie et al., 2013; Li et al., 2013; Feng et al., 2016). For example, a maximum $O_3$
concentration of 286 ppb has been observed in urban plumes from Beijing (Wang et al.,
2006). Chen et al. (2015) have reported that the average maximum daily $[O_3]$ exceed 150 μg
$m^{-3}$ in the summer of 2015 at most of monitoring sites in Beijing. Wu et al. (2016) have also
shown that, during summertime of 2015 in Beijing, the average $O_3$ concentration in the
afternoon is 163.2 μg $m^{-3}$, and the frequency of the $O_3$ exceedance with hourly $[O_3]$
exceeding 200 μg $m^{-3}$ is 31.8%. In addition, Cheng et al. (2016) have demonstrated an
increasing trend of daily maximum 1-h $[O_3]$ from 2004 to 2015 in Beijing, and Ma et al.
(2016) have reported significant increase of surface $O_3$ at a rural site in NCP. In PRD region,
the annual average near-surface $O_3$ level has been reported to increase from 24 ppbv in 2006
to 29 ppbv in 2009, and the maximum 1-h $[O_3]$ can be up to 150 ~ 200 ppb in the summer
and fall (Ou et al., 2016, EST). Numerous studies have been performed to examine the severe
$O_3$ pollution in China, but primarily confined to mega-cities or industrial complexes. Few
studies have been conducted in whole Eastern China to investigate the $O_3$ pollution situation
and formation.

The China's Ministry of Environmental Protection (China MEP) has commenced to

release real-time hourly observations of pollutants, including $O_3$, $NO_2$, CO, $SO_2$, $PM_{2.5}$, and
$PM_{10}$ (particulate matter with aerodynamic diameter less than 2.5 and 10 μm, respectively)
since 2013. In Eastern China, there are 65 cities with air pollutants observations in 2013
during summertime, mainly concentrated in Beijing-Tianjin-Hebei (BTH), YRD, and PRD




(Figure 1). In 2015, a total of 223 cities have air pollutants observation in Eastern China,
providing a good opportunity to explore the $O_3$ pollution distributions. Therefore, in the
present study, the $O_3$ pollution situation in 2015 is first analyzed from April to September
when [$O_3$] are high in Eastern China. A high $O_3$ episode occurred in Eastern China in 2015 is
simulated using the WRF-CHEM model to evaluate the $O_3$ formation from biogenic and
various anthropogenic sources. The WRF-CHEM model configuration and methodology are
described in Section 2. Data analysis and model results are presented in Section 3, and
conclusions and discussions are given in Section 4.

**2      Model and Methodology**
**2.1    WRF-CHEM Model and Configurations**

In the present study, we use a specific version of the WRF-CHEM model (Grell et al.,

2005) to investigate the $O_3$ formation in Eastern China. The model is developed by Li et al.
(2010; 2011a, b; 2012) at the Molina Center for Energy and the Environment, including a
new flexible gas phase chemical module and the CMAQ/Models3 aerosol module developed
by US EPA (Binkowski and Roselle, 2003). The wet deposition of chemical species is
calculated using the method in the CMAQ module and the dry deposition parameterization
follows Wesely (1989). The FTUV is used to calculate photolysis rates (Tie et al., 2003; Li et
al., 2005), considering the impacts of aerosols and clouds on the photochemistry (Li et al.,
2011b). The ISORROPIA Version 1.7 is used to calculate the inorganic aerosols (Nenes et al.,
1998). The secondary organic aerosol (SOA) is predicted using a non-traditional SOA
module, including the volatility basis-set (VBS) modeling approach and SOA contributions
from glyoxal and methylglyoxal. Detailed information about the WRF-CHEM model can be
found in Li et al. (2010; 2011a, b; 2012).

A high $O_3$ pollution episode from 22 to 28 May 2015 in Eastern China is simulated





using the WRF-CHEM model. The simulation domain is shown in Figure 1. Detailed model
configurations are given in Table 1. For discussion convenience, Eastern China is divided
into four sections: 1) the Northeast China (including Heilongjiang, Jilin, Liaoning, and the
east part of Inner Mongolia, hereafter referred to as NEC), 2) the North China Plain and
surrounding areas (including Beijing, Tianjin, Hebei, Shandong, Henan, Shanxi, and the
north part of Jiangsu and Anhui, hereafter referred to as NCPs), 3) the YRD and surrounding
areas (including the south part of Jiangsu and Anhui, Shanghai, Zhejiang, and Hubei,
hereafter referred to as YRDs), and 4) the PRD and surrounding areas (including Fujian,
Jiangxi, Hunan, Guangxi, and Guangdong, hereafter referred to as PRDs) (shown in
Supplementary Information (SI), SI-Figure 1).
**2.2    Statistical Methods for Comparisons**
We use the mean bias (*MB*) and the index of agreement (*IOA*) to assess the WRF-
CHEM model performance in simulating air pollutants against measurements.
$$MB = \frac{1}{N}\sum_{i=1}^{N}(P_i - O_i)$$
$$IOA = 1 - \frac{\sum_{i=1}^{N}(P_i - O_i)^2}{\sum_{i=1}^{N}(|P_i - \bar{O}| + |O_i - \bar{O}|)^2}$$
where $P_i$ and $O_i$ are the calculated and observed pollutant concentrations, respectively. $N$ is
the total number of the predictions used for comparisons, and $\bar{P}$ and $\bar{O}$ represents the average
of the prediction and observation, respectively. The *IOA* ranges from 0 to 1, with 1 showing
perfect agreement of the prediction with the observation.
**2.3    Air Pollutants Measurements**
The hourly near-surface CO, $NO_2$, $SO_2$, and $PM_{2.5}$ mass concentrations from April to
September 2015 in Eastern China are released by China MEP, and can be downloaded from
the website http://www.aqistudy.cn/.

**3    Results and Discussions**



### 3.1 O₃ pollution in Eastern China

**3.1   O$_3$ pollution in Eastern China**
Continuous deterioration of air qualtiy in China has engendered the implementation of
"Atmospheric Pollution Prevention and Control Action Plan" (hereafter referred to as
APPCAP), released by Chinese State Council in September 2013 to reduce $PM_{2.5}$ by up to 25%
by 2017 relative to 2012 levels. Therefore, variations of air pollutants from 2013 to 2015
demonstrate the mitigation effects of implementation of the APPCAP on the air quality to a
considerable degree. A total of 65 cities, with 427 monitoring sites, have air pollutants
observations from 2013 to 2015 during April to September in Eastern China (Figure 1).
Considering the occurrence of high $[O_3]$ in the afternoon (12:00 – 18:00 Beijing Time (BJT)),
Table 2 provides the average concentrations of air pollutants in the afternoon from April to
September in the 65 cites of Eastern China in 2013 and 2015. Apparently, implementation of
the APPCAP has decreased the mass concentrations of CO, $SO_2$, $NO_2$, and $PM_{2.5}$ in Eastern
China, particularly with regard to $SO_2$, with a reduction of close to 40% from 2013 to 2015.
The $[O_3]$ however exhibit an increasing trend, enhanced by 9.9% from 2013 to 2015.
Additionally, if the $O_3$ exceedance is defined as hourly $[O_3]$ exceeding 200 $\mu g\ m^{-3}$ (the
second grade of National Ambient Air Quality Standards in China), the $O_3$ exceedance
frequency in the afternoon has increased from 5.2% in 2013 to 6.8% in 2015, enhanced by
about 31.5%. There are several possible reasons for the $O_3$ pollution deterioration in Eastern
China since implementation of the APPCAP. Firstly, if the $O_3$ production regime in Eastern
China is $NO_x$-sensitive, the decrease of $NO_x$ due to implementation of the APPCAP likely
enhances the $O_3$ formation. Secondly, mitigation of $PM_{2.5}$ or aerosols directly or indirectly
increases the photolysis rates and expedites the $O_3$ formation. Thirdly, increasing
transportation activities enhances the emissions of VOCs and semi-VOCs, facilitating the $O_3$
formation. In addition, variability of meteorological situations also leads to the $[O_3]$



fluctuation. Hence, implementation of the APPCAP does not help mitigate [$O_3$], and
unfortunately, severe $O_3$ pollutions have been looming in Eastern China.

In 2015, $O_3$ observations have been performed in 223 cities with 1064 monitoring sites

in Eastern China, which are used to analyze the $O_3$ pollution situation from April to
September. For comparisons, Figure 2 shows the distribution of observed maximum 1-h [$O_3$]
in Mainland China from April to September in 2015. The cities with the maximum 1-h [$O_3$]
exceeding 300 μg m$^{-3}$ are mainly concentrated in NCPs, YRDs, and PRD. In Eastern China,
there are only two cities with the maximum 1-h [$O_3$] less than 200 μg m$^{-3}$. About 28% of
cities have observed more than 400 μg m$^{-3}$ [$O_3$] (about 200 ppb), showing widespread $O_3$
pollution in Eastern China. Furthermore, it is worth to note that the observed maximum 1-h
[$O_3$] in six cites exceed 800 μg m$^{-3}$ (about 400 ppb), in a very dangerous level.

Figure 3 presents the distribution of average daily maximum 1-h [$O_3$] in Mainland

China from April to September 2015. The average daily maximum 1-h [$O_3$] are more than
120 μg m$^{-3}$ in more than 95% of the cities, and 160 μg m$^{-3}$ in 46% of the cities in Eastern
China. Particularly, there are seven cities with the average daily maximum 1-h [$O_3$]
exceeding 200 μg m$^{-3}$ during six months. Figure 4 and 5 show the distributions of exceedance
days with the maximum 1-h [$O_3$] exceeding 160 and 200 μg m$^{-3}$ in Mainland China from
April to September 2015, respectively. There are more than 60 days with the maximum 1-h
[$O_3$] exceeding 160 μg m$^{-3}$ in 114 cities, and even more than 90 days in 62 cites in Eastern
China from April to September. The 1-h [$O_3$] of 200 μg m$^{-3}$ have been exceeded on over 10%
of days in 129 cities, and on 30% of days in 38 cities (Figure 5). Hence, persistent $O_3$
pollution has occurred in Eastern China from April to September in 2015.

Furthermore, in the urban PBL, high [$O_3$] generally take place under calm or stable

circumstances with strong solar radiation. From April to September, the East Asian summer
monsoon influences Eastern China, causing intensified precipitation which inhibits the high



O₃ formation through washing out O₃ precursors and decreasing photolysis rates. So if
excluding rainy days in the analysis, the O₃ pollution becomes more severe in Eastern China.
For example, in Beijing, there are 54 rainy days and 65 days with the maximum 1-h [O₃]
exceeding 200 μg m⁻³ from May to August in 2015. If it does not rain in Beijing, the
occurrence possibility of the maximum 1-h [O₃] exceeding 200 μg m⁻³ is around 94%,
showing severe and persistent O₃ pollution.
**3.2   Model Performance**

The hourly measurements of O₃ and NO₂ in Eastern China are used to validate the

WRF-CHEM model simulations. Figure 6 presents the distributions of calculated and
observed near-surface [O₃] along with the simulated wind fields at 15:00 BJT from 22 to 27
May 2015. On May 22, Eastern China is influenced by the subtropical high whose center
locates over the Yellow sea. The east winds in the south of the high transport humid air into
PRDs, causing rainfall weather that substantially decreases [O₃]. The WRF-CHEM model
well reproduces the observed low [O₃] in the south of PRDs. In NCPs and YRDs, calm winds,
clear sky, and high temperature, induced by the high, facilitate the O₃ formation, and the
simulated [O₃] generally exceed 160 μg m⁻³, which is consistent with the observations. On
May 23, the subtropical high moves northward, also causing the rainfall belt in the south of
PRDs to extend northward. The simulated O₃ pollution in NCPs is deteriorated and also
extended to NEC, in good agreement with the measurements. From May 24 to 25, the
stagnant subtropical high continuously deteriorates the O₃ pollution in Eastern China. The
simulated and observed O₃ pollution on May 25 is widespread almost in Eastern China, and
the Northwest China also experiences high O₃ pollution. On May 26 and 27, the subtropical
high moves northward again and the rainfall belt has advanced to the south of NCPs. The
simulated and observed [O₃] in the north of NCPs and NEC are still high, but PRDs and
YRDs, the [O₃] have been significantly decreased due to precipitation. Generally, the





simulated $O_3$ spatial patterns are consistent with observations, but the model underestimation
or overestimation still exists. For example, the model remarkably overestimates the observed
$[O_3]$ on May 24, and also cannot well reproduce the high $[O_3]$ on May 25 in PRD. There are
several reasons for the model biases in simulating $[O_3]$ distribution. Firstly, the
meteorological situations play a key role in air pollution simulations (Bei et al., 2010, 2012),
determining the formation, transformation, diffusion, transport, and removal of the air
pollutants. Therefore uncertainties in meteorological fields simulations significantly influence
the air pollutants simulations. On May 24, the model fails to predict the rainy or overcast
weather, leading to remarked overestimation of $[O_3]$ in PRD. Secondly, the 10 km horizontal
resolution is used in simulations, which cannot resolve well cumulus clouds. The model
overestimates the $[O_3]$ observed in some cities with $[O_3]$ much lower than their surrounding
cities, which is primarily caused by the model failure in resolving convections. Thirdly, the
fast changes in emissions are not reflected in the emissions inventories used in the present
study.

Figure 7 provides the diurnal profiles of calculated and observed near-surface $[O_3]$

averaged over the ambient monitoring sites in provinces and municipalities in Eastern China
during the episode. The model reasonably well reproduces the temporal variations of surface
$[O_3]$ compared to observations, e.g., peak $[O_3]$ in the afternoon due to active photochemistry
and low $[O_3]$ during nighttime caused by the $NO_x$ titration. Three provinces in NEC, Jilin,
Liaoning, and Inner Mongolia, are apparently impacted by the trans-boundary transport from
NCPs when the south winds are prevailing (Figure 6). So the uncertainties of wind field
simulations constitute one of the most important reasons for the model biases in modeling
$[O_3]$ in these three provinces. The model underestimates considerably the observed $[O_3]$ in
the three provinces (Figures 7a, c, d), with $MB$s exceeding 19 $\mu g\ m^{-3}$. The model generally
exhibits good performance in simulating $[O_3]$ variations in the provinces of NCPs (Figures



7e-l) with *IOA*s exceeding 0.90, but is subject to underestimate the observations, particularly
in Beijing which is also significantly influenced by the trans-boundary transport (Wu et al.,
2016). In YRDs, the model cannot well predict the observed [$O_3$] in Shanghai, which is
affected by the sea breeze when the large-scale wind fields are weak. In general, however,
current numerical weather prediction models, even in research mode, still have difficulties in
producing the location, timing, depth, and intensity of the sea-breeze front (Banta et al.,
2005). The model reasonably predicts the [$O_3$] variations compared to measurements in PRDs
(Figures 7p-t) with *IOA*s more than 0.7, but overestimates the observed [$O_3$] with *MB*s
varying from 3.8 to 16.7 μg m$^{-3}$, showing model biases in modeling precipitation processes.
The comparisons of simulated vs. observed distributions and temporal variations of
$NO_2$ mass concentrations ([$NO_2$]) are shown in Supplementary Information (SI, SI-Figures 2
and 3). The simulated high near-surface [$NO_2$] are mainly concentrated in NCP, YRD, and
PRD, which is generally consistent with the measurements. The model also reasonably yields
temporal variations of [$NO_2$] compared to measurements, but the simulations of [$NO_2$] are
not as good as those of [$O_3$], and the *IOA*s in Liaoning, Tianjin, and Shanghai are lower than
0.5. The difference between simulations and observations are frequently rather large during
nighttime, which perhaps caused by the model biases in modeling nighttime PBL or the
complexity of nighttime chemistry. In general, the calculated distributions and variations of
[$O_3$] and [$NO_2$] are consistent with the corresponding observations, showing that the
simulations of meteorological fields and emissions inventories are reasonable, providing the
base for sensitivity studies.
**3.3   Sensitivity Studies**
$O_3$ formation in the PBL is a complicated nonlinear process, depending on its
precursors of $NO_x$ and VOCs from biogenic and various anthropogenic sources. It is
imperative to evaluate the $O_3$ contribution from various sources for devising the $O_3$ control



strategy. Rapid growth of industries, transportation, and urbanization has caused increasing
emissions of $NO_x$ and VOCs in Eastern China (e.g., Zhang et al., 2009; Huang et al., 2011;
Wang et al., 2012; Wang et al., 2013; Yang et al., 2015). Numerous studies have also
demonstrated that biogenic VOCs, such as isoprene and monoterpenes, play a considerable
role in the $O_3$ formation in the PBL (e.g., Chameides et al., 1988; Tao et al., 2003; Li et al.,
2007; 2014). Therefore, sensitivity studies are used to evaluate the $O_3$ contributions of
biogenic, industry, residential, and transportation sources in Eastern China, respectively. It is
worth to note that emissions of power plants are directly associated with residential living
and industrial activities. So in the study, 75% of emissions from power plants are assigned to
the industry source and the rest are assigned to the residential source according to the ratio of
the power consumption used in industrial activities to residential living (Wang et al., 2016).

The factor separation approach (FSA) is used to evaluate the contribution of some

emission source to the $O_3$ concentration by differentiating two model simulations: one with
all emissions sources and the other without some emission source. Therefore, except the
control simulations with all emissions, additional four sensitivity simulations are performed,
in which the biogenic, industry, residential, and transportation emissions are excluded,
respectively, to assess their corresponding contributions to the $O_3$ formation in Eastern China.

Figure 8 shows the contribution of near-surface [$O_3$] averaged in the afternoon during

the whole episode from industry, residential, transportation, and biogenic emissions. The
industry source plays a more important role in the $O_3$ formation than the rest three sources,
with the $O_3$ contribution of 10 ~ 50 $\mu g\ m^{-3}$ in the afternoon in Eastern China. In highly
industrialized areas, such as Hebei, Tianjin, Shandong, Zhejiang, et al., the $O_3$ contribution of
the industry source exceeds 30 $\mu g\ m^{-3}$. The residential source is not important in the $O_3$
formation, and contributes about 2 ~ 15 $\mu g\ m^{-3}$ $O_3$ generally. The transportation source plays
a considerable role in the $O_3$ formation, accounting for about 5 ~ 30 $\mu g\ m^{-3}$ $O_3$ in Eastern



China. The $O_3$ enhancement due to biogenic emissions is mainly concentrated in NCPs and
PRDs, particularly in PRDs, with the $O_3$ contribution of around $5 \sim 50$ µg m$^{-3}$.

In order to further evaluate the contribution of various sources to the [$O_3$], the hourly

near-surface [$O_3$] in the control simulation are first subdivided into 16 bins with the interval
of 20 µg m$^{-3}$. [$O_3$] in the control and sensitivity simulations as the bin [$O_3$] are assembled
respectively, and an average of [$O_3$] in each bin are calculated. Figures 9 shows the
contributions of various emissions sources to [$O_3$] in the four sections of Eastern China
during the episode. The industry emission plays the most important role in the $O_3$ formation,
and is the culprit of the high $O_3$ pollution. When the [$O_3$] in the control simulation are less
than 100 µg m$^{-3}$, the industry source generally decreases [$O_3$]. However, when the simulated
[$O_3$] are more than around 200 µg m$^{-3}$, the $O_3$ contribution from the industry emissions
generally exceeds 50 µg m$^{-3}$, and when the simulated [$O_3$] are more than 300 µg m$^{-3}$, the
industrial $O_3$ contribution can be up to 100 µg m$^{-3}$, constituting one third of the [$O_3$]. The $O_3$
contribution from the residential source is not significant, generally less than 20 µg m$^{-3}$. The
transportation source plays the second most important role in the $O_3$ formation in NEC, NCPs,
and YRDs, but its $O_3$ contribution is much less than that from the industry source when the
simulated [$O_3$] are more than 150 µg m$^{-3}$. VOCs from the biogenic source generally enhance
the $O_3$ formation, providing a background $O_3$ source. The biogenic source contributes about
$10 \sim 50$ µg m$^{-3}$ $O_3$ when simulated [$O_3$] are more than 150 µg m$^{-3}$ in NEC, NCPs, and YRDs.
However, in PRDs, the biogenic emissions constitute the second most important $O_3$ source,
with the $O_3$ contribution exceeding 50 µg m$^{-3}$ when simulated [$O_3$] are more than 250 µg m$^{-3}$.
Apparently, controlling the industry emissions can substantially mitigate the severer $O_3$
pollution in Eastern China. If the industry emissions are not considered in model simulations,
on average, the [$O_3$] are generally not more than 200 µg m$^{-3}$ in NEC, YRDs, and PRDs, but
still can exceed 160 µg m$^{-3}$. In addition, excluding the industry source in NCPs does not





mitigate [$O_3$] as remarkably as in the other regions, indicating that other emission sources
also play an important role in the $O_3$ formation. Although the transportation emission is the
second most important $O_3$ source in NEC, NCPs, and PRDs, its $O_3$ contribution is much less
than that from the industry source.
Another three sensitivity studies are conducted to further explore the high $O_3$ formation
in Eastern China, in which only the industry, residential, and transportation source is
considered, respectively. It is worth to note that biogenic emissions are included in all the
three sensitivity simulations considering that the biogenic emissions provide natural $O_3$
precursors and cannot be anthropogenically controlled. Figure 10 presents the $O_3$
contributions from individual anthropogenic source averaged in the afternoon during the
whole episode in the four sections of Eastern China. If only the industry source is considered
or the residential and transportation sources are excluded in the simulation, Eastern China
still experiences high $O_3$ pollution. The $O_3$ contribution of the residential and transportation
sources are less than 60 µg m$^{-3}$ on average, further showing the important role of the industry
source in the $O_3$ pollution. When the industry and residential sources are not considered in
the simulation, the transportation source still causes the simulated [$O_3$] to exceed 160 µg m$^{-3}$,
particularly in NCPs. Taking into consideration the very fast increase of vehicles in China
recently (X. Wu et al., 2016), the transportation source increasingly constitutes a more
important $O_3$ source, particularly when the industry source is under control. Apparently,
when the industry and transportation sources are excluded or only residential source is
included, the high $O_3$ pollution is significantly mitigated and the simulated [$O_3$] are less than
160 µg m$^{-3}$ on average. Figure 11 provides the distribution of the [$O_3$] averaged during the
peak time on May 25 when the most serous $O_3$ pollution occurs during the simulated episode.
When only the industry emissions are considered, the $O_3$ pollution is mitigated considerably
in Eastern China, but still widespread in NCPs and PRDs. If only considering the



transportation source, the $O_3$ pollution still occurs in NCPs, with the $[O_3]$ exceeding 160 μg
$m^{-3}$. When the industry and transportation sources are excluded, the $O_3$ pollution is generally
under control. Hence, reducing the emissions from industry and transportation is a key to
mitigate $O_3$ pollution in Eastern China.

**4     Summary and Conclusions**

In the present study, air pollutants observations, released by China MEP, have been

analyzed to explore the $O_3$ pollution situation in Eastern China. Analysis of air pollutants
observations in 66 cities from 2013 to 2015 have shown that, although implementation of the
APPCAP has considerably decreased the CO, $SO_2$, $NO_2$, and $PM_{2.5}$ mass concentrations from
April to September in Eastern China, the $[O_3]$ have increased by 9.2% and the frequency of
$O_3$ exceedance with hourly $[O_3]$ exceeding 200 μg $m^{-3}$ has increased by about 25% in the
afternoon. Mitigation of $NO_x$ and $PM_{2.5}$ due to implementation of the APPCAP, increasing
transportation activities, or variability of meteorological situations perhaps contributes to the
deterioration of the $O_3$ pollution in Eastern China.

$O_3$ observations from April to September in 2015 have shown that Eastern China has

experienced widespread and persistent $O_3$ pollution. Only two cities in Eastern China have
observed the maximum 1-h $[O_3]$ less than 200 μg $m^{-3}$. Over 25% of cities have observed the
maximum 1-h $[O_3]$ exceeding 400 μg $m^{-3}$, particularly more than 800 μg $m^{-3}$ $[O_3]$ have been
observed in six cities in Eastern China. The average daily maximum 1-h $[O_3]$ from April to
September exceed 160 μg $m^{-3}$ in 45% of cities in Eastern China, and the 1-h $[O_3]$ of 200 μg
$m^{-3}$ have been exceeded on over 10% of days from April to September in 129 cities, and on
40% of days in 10 cities.

A widespread and severe $O_3$ pollution episode from 22 to 28 May 2015 in Eastern

China has been simulated using the WRF-CHEM model. The model generally simulates



reasonably well the temporal variations and spatial distributions of near-surface $[O_3]$, but the
uncertainties of meteorological fields or emission inventories still cause model
overestimation or underestimation. The model performs reasonably in simulating $NO_2$, but
the model biases are rather large during nighttime.

FSA is utilized to assess the $O_3$ contribution of biogenic and various anthropogenic

sources. Sensitivity studies have shown that the industry source plays the most important role
in the $O_3$ pollution formation. When the simulated $[O_3]$ are more than around 200 $\mu g\ m^{-3}$, the
$O_3$ contribution from the industry emissions generally exceeds 50 $\mu g\ m^{-3}$ in Eastern China,
particularly when the simulated $[O_3]$ exceed 300 $\mu g\ m^{-3}$, the industrial $O_3$ contribution
constitutes one third of the $[O_3]$. The transportation emission is the second most important $O_3$
source in NEC, YRDs, and PRDs, but its $O_3$ contribution is much less than that from the
industry source when the simulated $[O_3]$ exceed 150 $\mu g\ m^{-3}$. The biogenic source plays a
more important role in $O_3$ formation than the transportation source in PRDs, with the $O_3$
contribution exceeding 50 $\mu g\ m^{-3}$ when simulated $[O_3]$ are more than 250 $\mu g\ m^{-3}$. In general,
the $O_3$ contribution from residential source is not significant.  Further sensitivity studies have
also indicated that if only considering the residential source or excluding the industry and
transportation sources in simulations, the $O_3$ pollution in Eastern China could be significantly
improved. Only the industry or transportation source still causes $O_3$ pollution, particularly
with regard to the industry source.

Widespread and persistent $O_3$ pollution poses adverse impacts on ecosystems and

human health. Considering the key role of the industry source in the high $O_3$ formation,
mitigation of the industry source becomes the top choice to improve the $O_3$ pollution in
Eastern China, particularly with regard to the VOCs emissions that are still not fully
considered in the current air pollutant control strategy. Rapid increase of vehicles also
enhances the VOCs and $NO_x$ emissions and the transportation source plays an increasingly



important role in the $O_3$ pollution. In addition, the rapid decrease of $PM_{2.5}$ due to
implementation of the APPCAP reduces the aerosol and cloud optical depth, which is subject
to enhance the $O_3$ formation by increasing the photolysis. Hence, stringent control strategies
of VOCs and $NO_x$ need to be designed comprehensively and implemented to avoid the
looming severe $O_3$ pollution in Eastern China.

Although the model performs generally well in simulating $O_3$ and $NO_2$ during a seven-

day $O_3$ pollution episode in Eastern China, uncertainties from meteorological fields
simulations and emissions inventory still cause model biases. Taking into consideration the
complexity of the $O_3$ formation and rapid changes of emissions inventories, further model
studies need to be performed to investigate the $O_3$ formation for supporting the design and
implementation of emission control strategies, based on the improved meteorological fields
simulations.

*Acknowledgements.* This work was supported by the National Natural Science Foundation of
China (No. 41275153) and by the "Strategic Priority Research Program" of the Chinese
Academy of Sciences,Grant No. XDB05060500. Guohui Li is also supported by the
"Hundred Talents Program" of the Chinese Academy of Sciences. Naifang Bei is supported
by the National Natural Science Foundation of China (No. 41275101). Wenting Dai is
supported by the National Natural Science Foundation of China (No. 41503117).



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

Contributions of Trans-boundary Transport to the Summertime Air Quality in Beijing,
China, Atmos. Chem. Phys. Discuss., 2016, 1-46, 10.5194/acp-2016-705, 2016.
Wu, X., Wu, Y., Zhang, S., Liu, H., Fu, L., and Hao, J.: Assessment of vehicle emission
programs in China during 1998-2013: Achievement, challlenges and implications,
Environ. Pollut., 2016, 214, 556-567, 2016.
Xing, J., Wang, S. X., Chatani, S., Zhang, C. Y., Wei, W., Hao, J. M., Klimont, Z., Cofala, J.,
and Amann, M.: Projections of air pollutant emissions and its impacts on regional air
quality in China in 2020, Atmos. Chem. Phys., 11, 3119-3136, 10.5194/acp-11-3119-
2011, 2011.

Xu J, Ma J Z, Zhang X L, et al. Measurements of ozone and its precursors in Beijing during
summertime: impact of urban plumes on ozone pollution in downwind rural areas, Atmos.
Chem. Phys., 2011, 11, 12241-12252.
Yang, X. F., Liu, H., Man, H. Y., and He, K. B.: Characterization of road freight
transportation and its impact on the national emission inventory in China, Atmos. Chem.
Phys., 15, 2105-2118, doi:10.5194/acp-15-2105-2015, 2015.
Zhang, Q., Streets, D. G., Carmichael, G. R., He, K. B., Huo, H., Kannari, A., Klimont, Z.,
Park, I. S., Reddy, S., Fu, J. S., Chen, D., Duan, L., Lei, Y., Wang, L. T., and Yao, Z. L.:
Asian emissions in 2006 for the NASA INTEX-B mission, Atmos. Chem. Phys., 9, 5131–
5153, doi:10.5194/acp-9-5131-2009, 2009.
Zhou, J. A., Ito, K., Lall, R., Lippmann, M., and Thurston, G.: Time-Series Analysis of
Mortality Effects of Fine Particulate Matter Components in Detroit and Seattle, Environ.
Health Persp., 119, 461-466, 10.1289/ehp.1002613, 2011.



Table 1 WRF-CHEM model configurations

| Regions | Eastern China |
| --- | --- |
| Simulation period | May 22 to 28, 2015 |
| Domain size | 350 × 350 |
| Domain center | 35°N, 114°E |
| Horizontal resolution | 10km × 10km |
| Vertical resolution | 35 vertical levels with a stretched vertical grid with spacing ranging from 30 m near the surface, to 500 m at 2.5 km and 1 km above 14 km |
| Microphysics scheme | WSM 6-class graupel scheme (Hong and Lim, 2006) |
| Boundary layer scheme | MYJ TKE scheme (Janjić, 2002) |
| Surface layer scheme | MYJ surface scheme (Janjić, 2002) |
| Land-surface scheme | Unified Noah land-surface model (Chen and Dudhia, 2001) |
| Longwave radiation scheme | Goddard longwave scheme (Chou and Suarez, 2001) |
| Shortwave radiation scheme | Goddard shortwave scheme (Chou and Suarez, 1999) |
| Meteorological boundary and initial conditions | NCEP 1°×1° reanalysis data |
| Chemical initial and boundary conditions | MOZART 6-hour output (Horowitz et al., 2003) |
| Anthropogenic emission inventory | SAPRC-99 chemical mechanism emissions (Zhang et al., 2009) |
| Biogenic emission inventory | MEGAN model developed by Guenther et al. (2006) |
| Model spin-up time | 28 hours |





Table 2 Observed hourly mass concentrations of pollutants averaged in the afternoon from
April to September 2013 and 2015 in 65 cities of Eastern China.

| Pollutants | CO (mg m$^{-3}$) | SO$_2$ (µg m$^{-3}$) | NO$_2$ (µg m$^{-3}$) | O$_3$ (µg m$^{-3}$) | PM$_{2.5}$ (µg m$^{-3}$) |
|---|---|---|---|---|---|
| 2013 | 1.05 | 24.8 | 27.7 | 100.5 | 46.9 |
| 2015 | 0.77 | 15.4 | 23.9 | 110.5 | 38.2 |
| Change (%) | -26.7 | -37.8 | -13.5 | +9.9 | -18.5 |




**Figure Captions**

Figure 1 WRF-CHEM simulation domain with topography. The filled circles represent centers of cities with ambient monitoring sites and the size of circles denotes the number of ambient monitoring sites of cities. The red and blue filled circles show the cities with air pollutants observations since 2013 and 2015, respectively.

Figure 2 Distribution of observed maximum 1-h $[O_3]$ in Mainland China from April to September 2015.

Figure 3 Distribution of average daily maximum 1-h $[O_3]$ in Mainland China from April to September 2015.

Figure 4 Distribution of days with the maximum 1-h $[O_3]$ exceeding 160 μg m$^{-3}$ in Mainland China from April to September 2015.

Figure 5 Distribution of days with the maximum 1-h $[O_3]$ exceeding 200 μg m$^{-3}$ in Mainland China from April to September 2015.

Figure 6 Pattern comparison of simulated vs. observed near-surface $O_3$ at 15:00 BJT from 22 to 27 May 2015. Colored circles: $O_3$ observations; color contour: $O_3$ simulations; black arrows: simulated surface winds.

Figure 7 Comparison of measured (black dots) and predicted (blue line) diurnal profiles of near-surface $O_3$ averaged over all ambient monitoring stations in provinces of Eastern China from 22 to 28 May 2015.

Figure 8 Distributions of the contribution to near-surface $[O_3]$ averaged in the afternoon during the whole episode from (a) industry, (b) residential, (c) transportation, and (d) biogenic emissions.

Figure 9 $O_3$ contributions of industry (red line), residential (brown line), transportation (blue line), and biogenic emissions (green line) in NEC, NCPs, YRDs, and PRDs, as a function of simulated $[O_3]$ in the control case.

Figure 10 $O_3$ contributions of industry alone (red line), residential (brown line), and transportation emissions (blue line) in NEC, NCPs, YRDs, and PRDs, as a function of simulated $[O_3]$ in the control case.

Figure 11 Distributions of the average $O_3$ concentration during peak time with (a) all anthropogenic emissions, (b) industry emissions alone, (c) residential emissions alone, and (d) transportation emissions alone on May 2015.



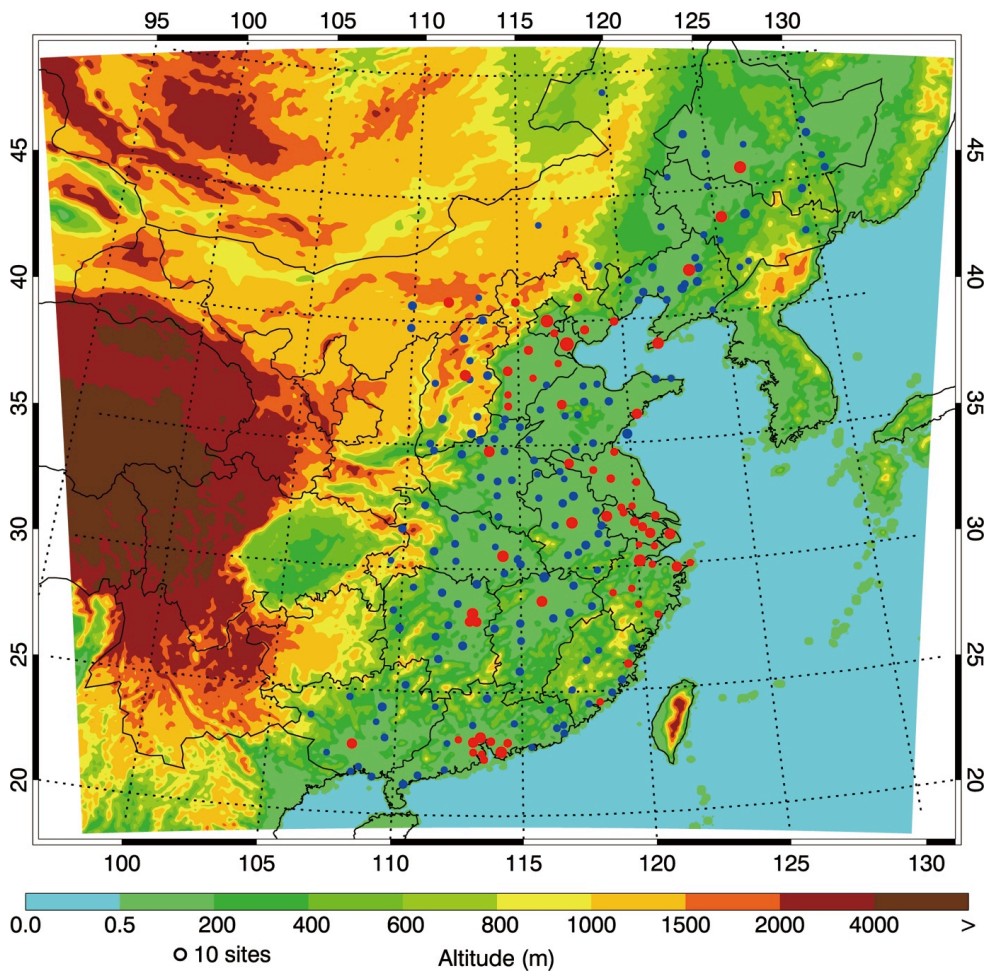

Figure 1 WRF-CHEM simulation domain with topography. The filled circles represent
centers of cities with ambient monitoring sites and the size of circles denotes the number of
ambient monitoring sites of cities. The red and blue filled circles show the cities with air
pollutants observations since 2013 and 2015, respectively.



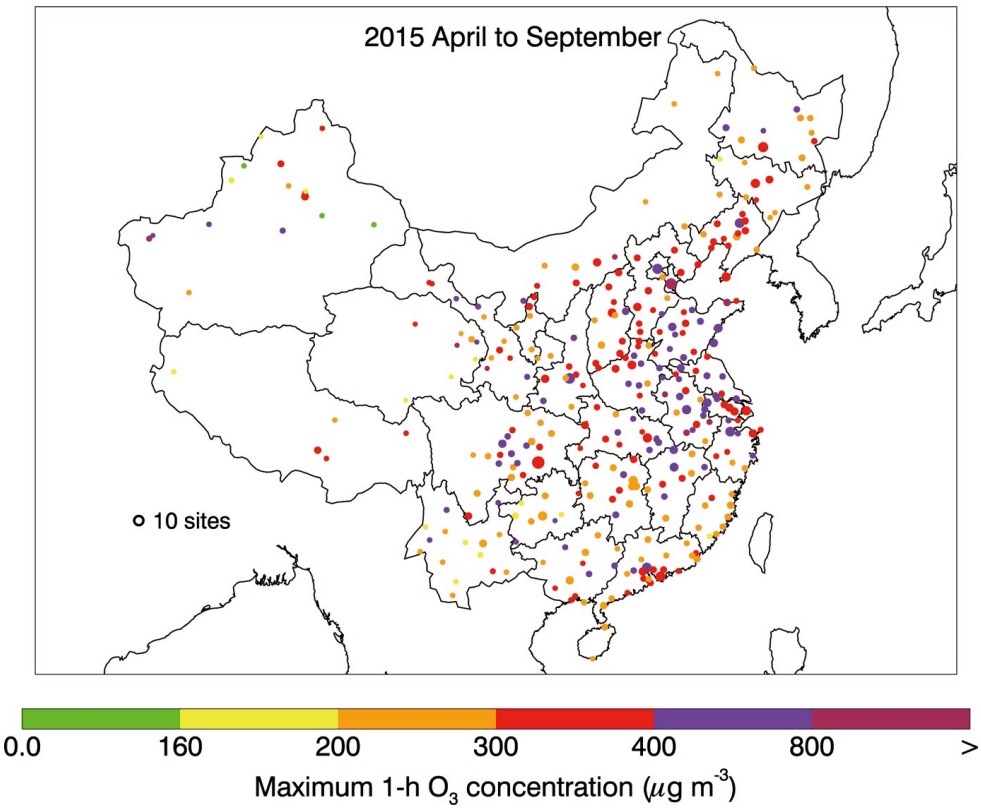

Figure 2 Distribution of observed maximum 1-h [O$_3$] in Mainland China from April to
September 2015.





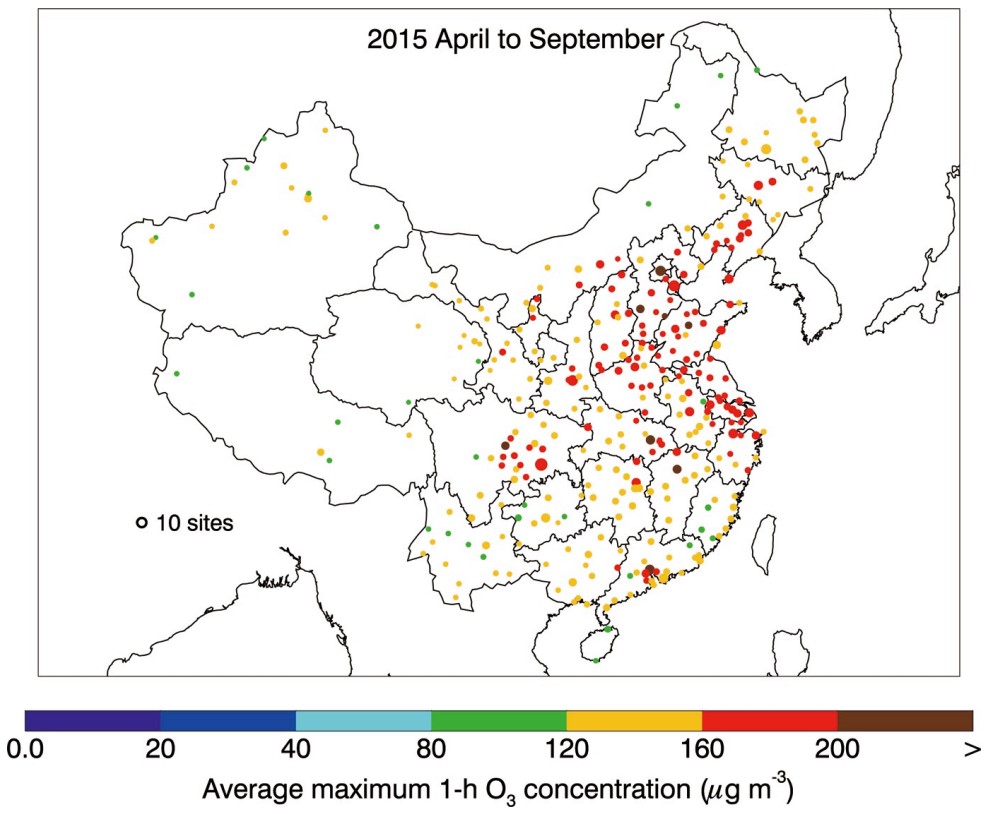

Figure 3 Distribution of average daily maximum 1-h [O₃] in Mainland China from April to
September 2015.





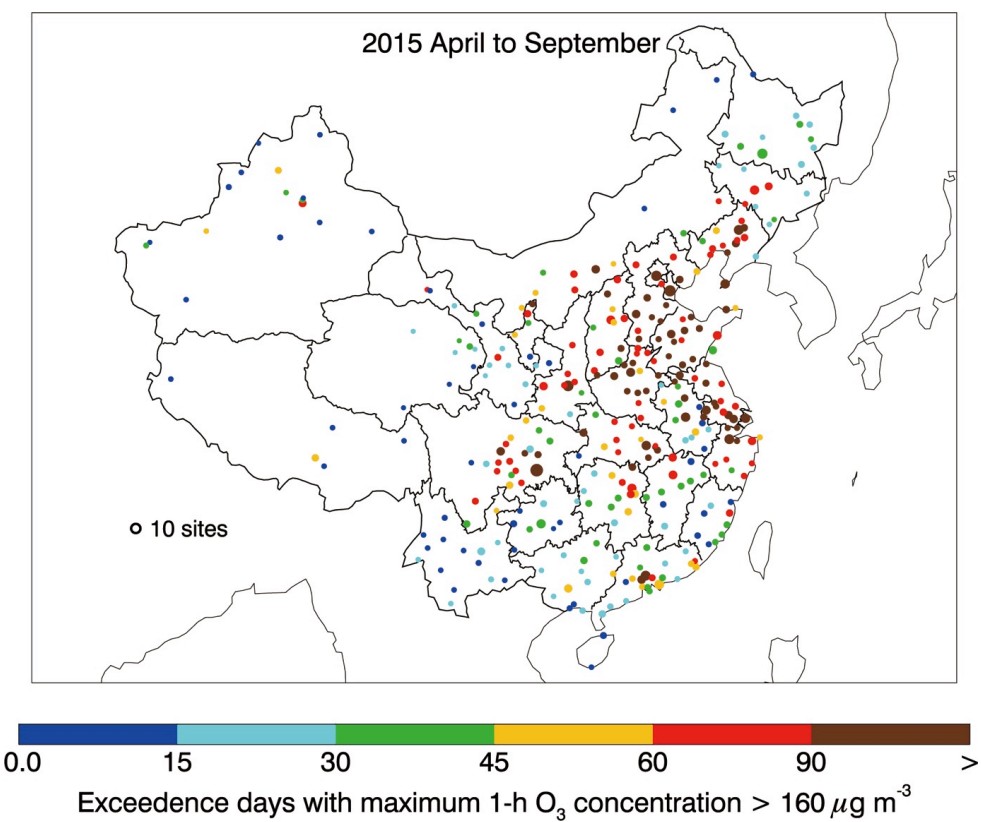

Figure 4 Distribution of days with the maximum 1-h [$O_3$] exceeding 160 µg m$^{-3}$ in Mainland
China from April to September 2015.



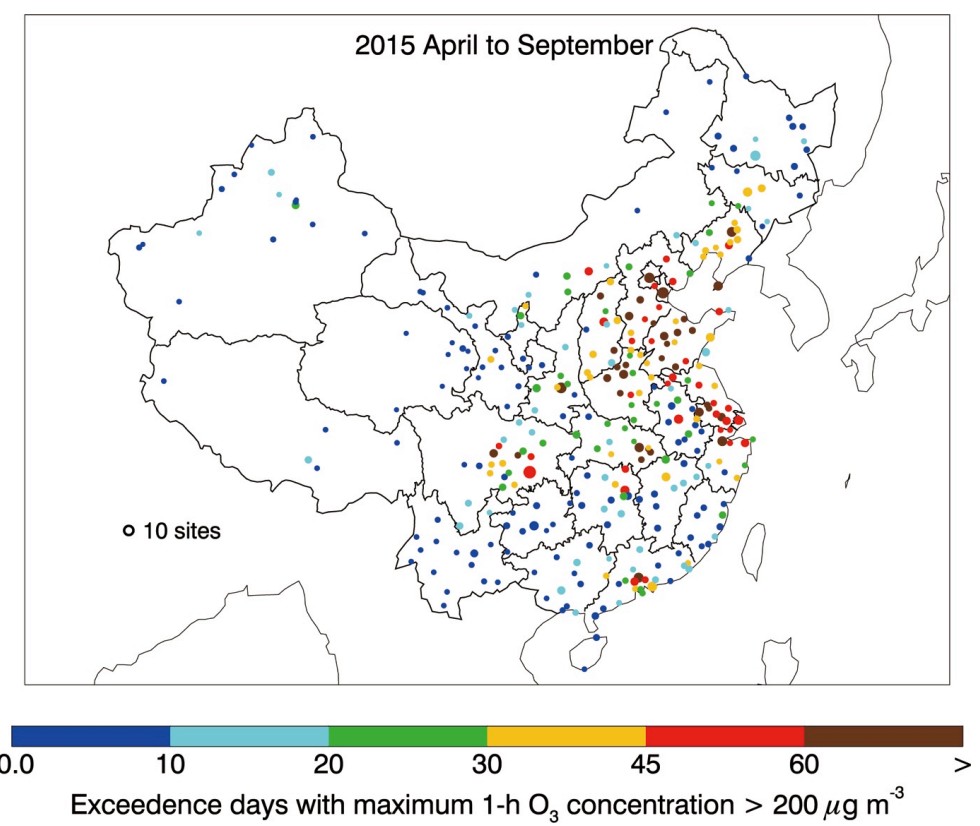

Figure 5 Distribution of days with the maximum 1-h [$O_3$] exceeding 200 μg m$^{-3}$ in Mainland
China from April to September 2015.





Figure 6 Pattern comparison of simulated vs. observed near-surface O₃ at 15:00 BJT from 22
to 27 May 2015. Colored circles: O₃ observations; color contour: O₃ simulations; black
arrows: simulated surface winds.





Figure 6 continued





Figure 6 continued







Figure 7 Comparison of measured (black dots) and predicted (blue line) diurnal profiles of
near-surface $O_3$ averaged over all ambient monitoring stations in provinces of Eastern China
from 22 to 28 May 2015.





Figure 7 continued





Figure 8 Distributions of the contribution to near-surface [$O_3$] averaged in the afternoon
during the whole episode from (a) industry, (b) residential, (c) transportation, and (d)
biogenic emissions.





Figure 8 continued



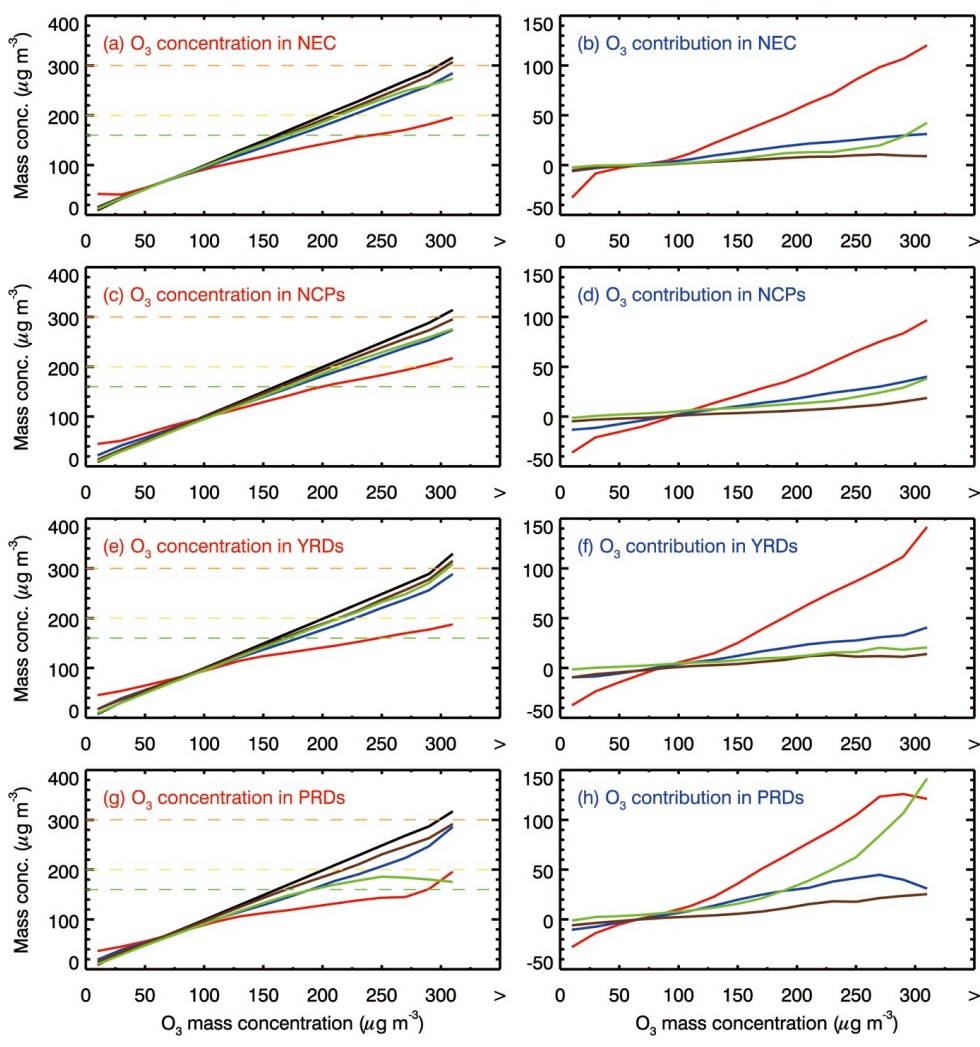

Figure 9 O$_3$ contributions of industry (red line), residential (brown line), transportation (blue
line), and biogenic emissions (green line) in NEC, NCPs, YRDs, and PRDs, as a function of
simulated [O$_3$] in the control case.



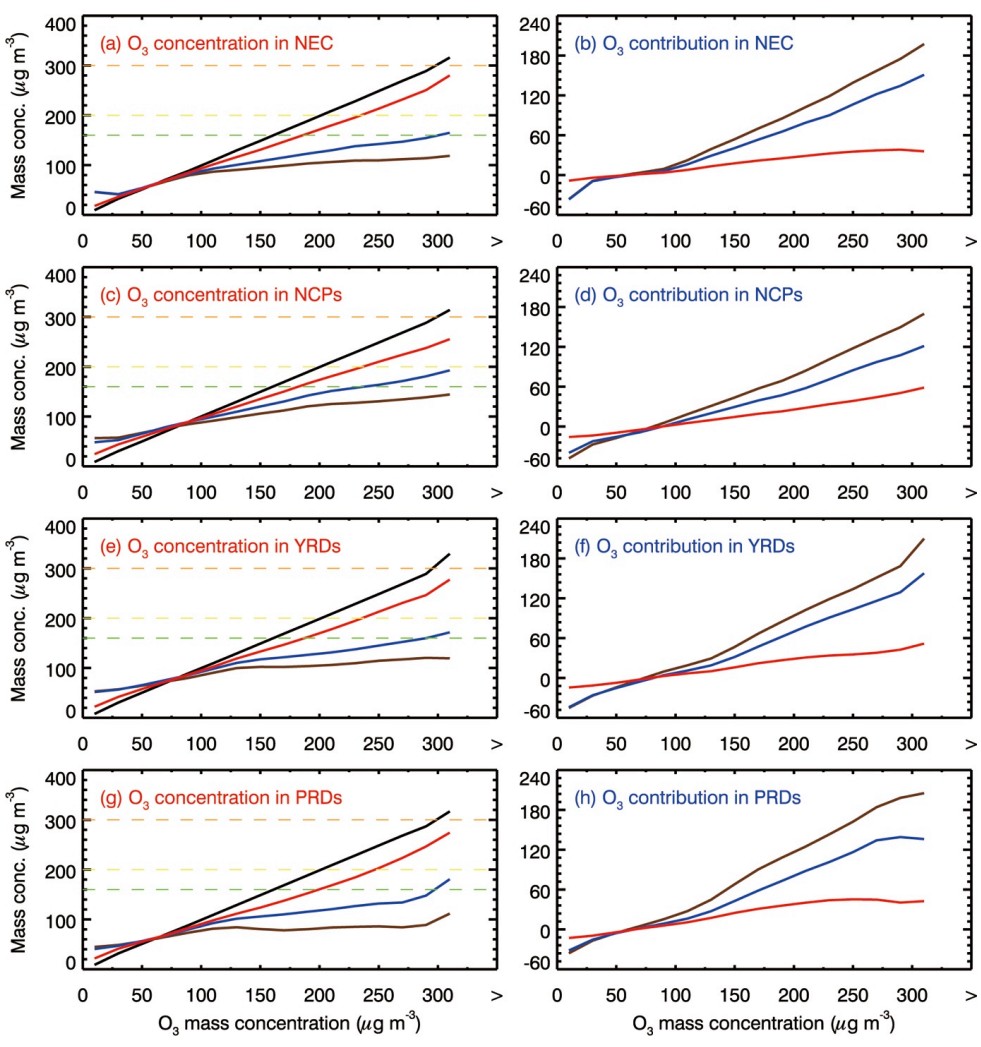

Figure 10 O$_3$ contributions of industry alone (red line), residential (brown line), and
transportation emissions (blue line) in NEC, NCPs, YRDs, and PRDs, as a function of
simulated [O$_3$] in the control case.



Figure 11 Distributions of the average $O_3$ concentration during peak time with (a) all
anthropogenic emissions, (b) industry emissions alone, (c) residential emissions alone, and (d)
transportation emissions alone on May 2015.





Figure 11 continued