# Peer review of "Widespread and Persistent Ozone Pollution in Eastern China during the Non-winter Season of 2015: Observations and Source Attributions"

_Atmospheric Chemistry and Physics, 2016_

## Referee Comment (RC1) · Anonymous Referee #2 · 5 Nov 2016

The manuscript presents observed distribution of surface O3 in April - Sep 2015 in East China. To interpret the causes of high O3 events, a modified version of WRFChem is used to simulated O3 distribution in 22-28 May 2015 and subsequent sensitivity experiments are conducted by turning off difference sources of emissions in the model. It is concluded that O3 pollution in Eastern China is widespread and persistent, and the most important cause for this is industrial emissions. The emission from transportation also has significant contribution to high O3 event, but the effect from residential emissions appears to be small.

The manuscript overall is sound, although it is questionable how representative the modeling analysis is. The title of this manuscript is not specific enough for describing what this paper is about. At the minimum, the title should add "observation and source attribution". It is a well known problem that pollution in China is widespread and

persistent, and there are many studies that have looked at this problem.

The manuscript needs a major revision before it can be considered for publication.

1. The title. suggesting to make changes to reflect the fact that the paper only looked the O3 in non-summer season and model simulation is done for only one case studies.

2. L50. 'photo' should be 'photon'.

3. L66-83. Are there any past modeling study if O3 in east China? If so, acknowledge it here.

4. section 2.1. how emission is set up for the model for both anthropogenic and biogenic sources? spatial resolution? boundary conditions? etc. is model spin up of 28 hours too short? need some description for the table 1.

5. Line 156. suggest to start a new paragraph, begin with "There are several".

6. Line 162. Suggest to add the following references that showed statistically, the meteorology has a significant impact on pollution.

Calkins, C., C. Ge, J. Wang, M. Anderson, K. Yang, 2016. Effects of meteorological conditions on sulfur dioxide air pollution in the North China Plain during winters of 2006-2015, Atmospheric Environment, 296-309.

7. Line. 156-160. If it is VOC limited, then decreasing of NOx will lead to increase of O3.

8. Section 3.2. Subtropic high pressure system is mentioned several times. Yet , all the figures show the surface wind only. Suggest to add either 500 hpa or 700 hpa geopotential heights into the map. see reference above.

9. section 3.3. how well the biogenic emission is represented, especially in high temperature conditions? suggest to add the following reference into the introduction and discussion. The agricultural section may contribute significant emission of NOx.

Oikawa P.Y. , C. Ge, J. Wang, J.R. Eberwein, L.L. Liang, L.A. Allsman, D.A. Grantz, and G.D. Jenerette, 2015. Unusually high soil nitrogen oxide emissions influence air quality in a high-temperature agricultural region, Nature Communications, 6, 8753.

10. Line 127-132. In equation for IOA, there is no $P^{\bar{}}\{bar\}$. Average of the prediction in what time/space?

11. Line 240-244. While for specific cases, perhaps this argument has some ground. However, on weekly to monthly basis, sea breeze should be well captured in the model. See breeze is added into the day to day wind vectors. Suggest to add the following reference in the discussion.

Wang, J., C. Ge, Z. Yang, E. J. Hyer, J. S. Reid, B.-N. Chew, M. Mahmud, Y. Zhang, and M. Zhang, 2013. Mesoscale modeling of smoke transport over the Southeast Asian Maritime Continent: interplay of sea breeze, trade wind, typhoon, and topography, Atmospheric Research , 122, 486-503.

12. Suggest to put Fig 2 - Fig. 5 into one figure, and note the difference in scale. Why use the maximum not second maximum 1-hr O3?

13. Figure 6 and other figures. there should be a legend for wind speed.

14. Fig. 7. There are places where observed peak O3 is well captured, well overestimated and well underestimated. I suggest short all the panels according to how well the peak O3 concentration is simulated by the model. Do you have any commonality where the peak O3 amount is not simulated well?

15. The only difference between captions for Fig. 9 and Fig. 10 is that caption 10 has a 'alone' after 'industry'. The caption should be clear about what we mean by 'alone' and not having word 'alone'. It is understood the O3 production is nonlinear.

---

## Referee Comment (RC2) · Anonymous Referee #1 · 8 Nov 2016

Ozone pollution is an emerging environmental issue in China, especially after the PM level started to decline. This paper analyzes surface measurements of ozone concentrations over 223 cites in Eastern China during 6 months in 2015 and quantitatively reveals the severity of ozone pollution during that period. A special version of WRF-Chem model developed by the authors is employed to investigate the relative contributions to the ozone formation from different sources, such as industry, transportation, residential and biogenic sources. The finding of industry sources as the culprit of the ozone pollution in Eastern China provides guidance on the future emission control strategy for police makers. Hence, I recommend accepting this paper by ACP after the authors address three minor comments below.

1. In Table 2, the comparison of pollutants between 2013 and 2015 shows that implementation of the emission control plan reduced NOx and PM concentrations but

resulted in a even worst O3 pollutions.
Such a phenomenon is quite interesting and should be highlighted in the abstract and conclusion. Does any satellite observation (such as OMI or TES on Aura ) capture such a change of ozone in Eastern China?

2. By turning off each emission source individually in the model, the authors tease out the role of each type of emission. One further question readers may have is what precursor species from each emission source are related to the ozone formation. It would be clearer if major VOC and NOx concentrations could be listed from each sectors in the emission dataset used by the WRF-Chem model.

3. Authors mentioned the possible uncertainty from the simulated meteorological conditions to explain the model biases in reproducing ozone distribution. Would a nudging of surface wind and temperature be helpful to minimize the influence of meteorology?

---

## Short Comment (SC1) · 20 Nov 2016

This is a nice paper reporting ozone pollution in China. I strongly encourage the authors to convert the unit for ozone mixing ratios from ug/m3 to ppbv. The latter (ppbv) is a more common unit used in the literature for ozone. By converting the unit to ppbv, the readers can quickly get a sense of how severe ozone pollution is in China, as compared to the other parts of the world.

Meiyun Lin (https://www.gfdl.noaa.gov/meiyun-lin-homepage/)

———————————————

---

## Author Comment (AC1) · 21 Jan 2017

**Reply to Anonymous Referee #2**

We thank the reviewer for the careful reading of the manuscript and helpful comments. We have revised the manuscript following the suggestion, as described below.

The manuscript presents observed distribution of surface $O_3$ in April - Sep 2015 in East China. To interpret the causes of high $O_3$ events, a modified version of WRF-Chem is used to simulate $O_3$ distribution in 22-28 May 2015 and subsequent sensitivity experiments are conducted by turning off difference sources of emissions in the model. It is concluded that $O_3$ pollution in Eastern China is widespread and persistent, and the most important cause for this is industrial emissions. The emission from transportation also has significant contribution to high $O_3$ event, but the effect from residential emissions appears to be small.

The manuscript overall is sound, although it is questionable how representative the modeling analysis is. The title of this manuscript is not specific enough for describing what this paper is about. At the minimum, the title should add "observation and source attribution". It is a well-known problem that pollution in China is widespread and persistent, and there are many studies that have looked at this problem. The manuscript needs a major revision before it can be considered for publication.

**1 Comment:** The title. Suggesting to make changes to reflect the fact that the paper only looked the $O_3$ in non-summer season and model simulation is done for only one case studies.

**Response:** We have changed the title as suggested to "Widespread and Persistent Ozone Pollution in Eastern China during the Non-winter Season of 2015: Observations and Source Attributions"

**2 Comment:** L50. "photo" should be "photon".

**Response:** We have changed the "photo" to "photon" in Section 1.

**3 Comment:** L66-83. Are there any past modeling study if O₃ in east China? If so, acknowledge it here.

**Response:** We have added a description of modeling study of O₃ in the Introduction as follows: "*Additonally, modeling studies have been performed to investigate the O₃ pollution in Eastern China (Wang et al., 2010; Liu et al., 2012; Situ et al., 2013; Huang et al., 2015). For example, Tie et al. (2013) have analyzed the characteristics of regional O₃ formation to explain the O₃ pollution in Shanghai and its surrounding area using the WRF-CHEM model. Using the observation-based chemical model, Xue et al. (2014) have provided insights into the ozone pollution in Beijing, Shanghai, and Guangzhou by analyzing the O₃ precursors and the potential impacts of heterogeneous chemistry.*"

**4 Comment:** Section 2.1. How emission is set up for the model for both anthropogenic and biogenic sources? Spatial resolution? Boundary conditions? etc. is model spin up of 28 hours too short? need some description for the table 1.

**Response:** We have added a paragraph in Section 2.1: "*The WRF-CHEM model adopts one grid with horizontal resolution of 10 km and 35 sigma levels in the vertical direction, and the grid cells used for the domain are 350 × 350 (Figure 1). The physical parameterizations include the microphysics scheme of Hong et al (Hong and Lim, 2006), the Mellor, Yamada, and Janjic (MYJ) turbulent kinetic energy (TKE) planetary boundary layer scheme (Janjić, 2002), the Unified Noah land-surface model (Chen and Dudhia, 2001), the Goddard long wave (Chou and Suarex, 2001) and shortwave parameterization (Chou and Suarex, 1999). The NCEP 1° × 1° reanalysis data are used to obtain the meteorological initial and boundary conditions, and the meteorological simulations are not nudged in the study. The chemical initial and boundary conditions are interpolated from the 6h output of MOZART (Horowitz et al., 2003). The spin-up time of the WRF-CHEM model is 28 hours, which is generally long enough for simulations considering that the initial and boundary conditions are adopted from MOZART, a*"

*global chemical transport model. The SAPRC-99 chemical mechanism is used in the present study. The anthropogenic emissions are developed by Zhang et al. (2009), including contributions from agriculture, industry, power generation, residential, and transportation sources. The biogenic emissions are calculated online using the MEGAN (Model of Emissions of Gases and Aerosol from Nature) model developed by Guenther et al (2006)."*

**5 Comment:** Line 156. Suggest to start a new paragraph, begin with "There are several".

**Response:** We have started a new paragraph beginning with "There are several" in Section 3.1.

**6 Comment:** Line 162. Suggest to add the following references that showed statistically, the meteorology has a significant impact on pollution.
Calkins, C., C. Ge, J. Wang, M. Anderson, K. Yang, 2016. Effects of meteorological conditions on sulfur dioxide air pollution in the North China Plain during winters of 2006-2015, Atmospheric Environment, 296-309.

**Response:** We have added the reference in Section 3.1.

**7 Comment:** Line. 156-160. If it is VOC limited, then decreasing of $NO_x$ will lead to increase of $O_3$.

**Response:** We have revised the sentence in Section 3.1 as "*If the $O_3$ production regime in eastern China is VOC-sensitive, the decrease of $NO_x$ due to implementation of the APPCAP likely enhances the $O_3$ formation.*"

**8 Comment:** Section 3.2. Subtropical high pressure system is mentioned several times. Yet , all the figures show the surface wind only. Suggest to add either 500 hpa or 700 hpa geopotential heights into the map. see reference above.

**Response:** We have added SI-Figure 2 in Supplemental Information (SI) material and described the 500hPa geopotential heights in Section 3.2 as follows: "*In order to interpret the effect of meteorological and synoptic conditions on the air quality in Eastern China, SI-Figure 2 presents the average geopotential height wind filed at 500 hPa from 22 to 27 May 2015. During the study episode, the NCPs and NEC are generally located behind the trough whose center is located between 120°E and 130°E. At the end of May, the main part of subtropical high at 500 hPa locates at the western Pacific, with the ridgeline moving around the 10°N -15°N. With the onset of summer monsoon, the subtropical high gradually moves northwards and affects Southern China, with more precipitation occurrence over YRDs and PRDs. Figure 6 presents the distributions of calculated and observed near-surface [O$_3$] along with the simulated wind fields at 15:00 BJT from 22 to 27 May 2015. On May 22, Eastern China is influenced by the high-pressure whose center locates over the Yellow sea, which is induced by the high level trough. The east winds in the south of the high transport humid air into PRDs, causing rainfall weather that substantially decreases [O$_3$]. The WRF-CHEM model well reproduces the observed low [O$_3$] in the south of PRDs. In NCPs and YRDs, calm winds, clear sky, and high temperature, induced by the high, facilitate the O$_3$ formation, and the simulated [O$_3$] generally exceed 160 μg m$^{-3}$, which is consistent with the observations. On May 23, the subtropical high moves northward, also causing the rainfall belt in the south of PRDs to extend northward. The simulated O$_3$ pollution in NCPs is deteriorated and also extended to NEC, in good agreement with the measurements. From May 24 to 25, the high pressure located at the Yellow sea continuously deteriorates the O$_3$ pollution in Eastern China. The simulated and observed O$_3$ pollution on May 25 is widespread almost in Eastern China, and the Northwest China also experiences high O$_3$ pollution. On May 26 and 27, the simulated and observed [O$_3$] in the north of NCPs and NEC are still high, but in PRDs and YRDs, the [O$_3$] have been significantly decreased due to the precipitation caused by the subtropical high and summer monsoon.*"

**9 Comment**: section 3.3. How well the biogenic emission is represented, especially in high temperature conditions? suggest to add the following reference into the introduction and discussion. The agricultural section may contribute significant emission of NO$_x$.

Oikawa P.Y. , C. Ge, J. Wang, J.R. Eberwein, L.L. Liang, L.A. Allsman, D.A. Grantz, and G.D. Jenerette, 2015. Unusually high soil nitrogen oxide emissions influence air quality in a high-temperature agricultural region, Nature Communications, 6, 8753.

**Response:** We have included the reference in the introduction (Section 1): "In addition, the agriculture has been proposed to have a large potential to produce $NO_x$ (Oikawa et al., 2015).", and in the discussion (Section 3.2): "Another possible reason for $NO_x$ biases in simulations is lack of consideration of the $NO_x$ emissions in the agricultural region, which has been proposed to generate high $NO_x$ emissions under high-temperature conditions (Oikawa et al., 2015)."

**10 Comment:** Line 127-132. In equation for IOA, there is no $P^{bar}$. Average of the prediction in what time/space?

**Response:** We have checked the equations and corrected them. The hourly predicted variables are averaged in the equations.

**11 Comment:** Line 240-244. While for specific cases, perhaps this argument has some ground. However, on weekly to monthly basis, sea breeze should be well captured in the model. See breeze is added into the day to day wind vectors. Suggest to add the following reference in the discussion.
Wang, J., C. Ge, Z. Yang, E. J. Hyer, J. S. Reid, B.-N. Chew, M. Mahmud, Y. Zhang, and M. Zhang, 2013. Mesoscale modeling of smoke transport over the Southeast Asian Maritime Continent: interplay of sea breeze, trade wind, typhoon, and topography, Atmospheric Research, 122, 486-503.

**Response:** We have added the reference in Section 3.2.

**12 Comment:** Suggest to put Fig.2 -Fig.5 into one figure, and note the difference in scale. Why use the maximum not second maximum 1-hr $O_3$?

**Response:** Considering that a total of 223 cities are shown on the map, if Figures 2~5 are put into one figure, the individual figure is too small to clearly show the observations. Furthermore, Figures 2~5 clarify the ozone pollution from different perspectives. In all figures, we use the maximum 1-hr $O_3$ as an indicator to analyze the $O_3$ pollution in Eastern China because the individual air quality index is based on it.

**13 Comment:** Figure 6 and other figures. There should be a legend for wind speed.

**Response:** We have added the legend for wind speed at the bottom of the color bar (right side) in Figure 6 and other figures.

**14 Comment:** Fig.7. There are places where observed peak $O_3$ is well captured, well overestimated and well underestimated. I suggest short all the panels according to how well the peak $O_3$ concentration is simulated by the model. Do you have any commonality where the peak $O_3$ amount is not simulated well?

**Response:** The panels are arranged according to the geographical positions of the provinces (for the north to the south). If sorting all the panels according to how well the peak $O_3$ concentration is simulated by the model, it is difficult for readers to follow the discussions in the manuscript. In addition, in general, the model performs best in simulating the peak $O_3$ amount in NCPs and also considerably well in NEC, YRDs and PRDs.

**15 Comment:** The only difference between captions for Fig.9 and Fig.10 is that caption 10 has a "alone" after "industry". The caption should be clear about what we mean by "alone" and not having word 'alone'. It is understood the $O_3$ production is nonlinear.

**Response:** We have revised the Figure 10 captions as "*$O_3$ contributions when only the industry (red line), residential (brown line), and transportation emissions (blue line) are considered in NEC, NCPs, YRDs, and PRDs, as a function of simulated [$O_3$] in the control case.*"

---

## Author Comment (AC2) · 21 Jan 2017

**Reply to Anonymous Referee #1**

We thank the reviewer for the careful reading of the manuscript and helpful comments. We have revised the manuscript following the suggestion, as described below.

Ozone pollution is an emerging environmental issue in China, especially after the PM level started to decline. This paper analyzes surface measurements of ozone concentrations over 223 cites in Eastern China during 6 months in 2015 and quantitatively reveals the severity of ozone pollution during that period. A special version of WRF-Chem model developed by the authors is employed to investigate the relative contributions to the ozone formation from different sources, such as industry, transportation, residential and biogenic sources. The finding of industry sources as the culprit of the ozone pollution in Eastern China provides guidance on the future emission control strategy for police makers. Hence, I recommend accepting this paper by ACP after the authors address three minor comments below.

**1 Comment:** In Table 2, the comparison of pollutants between 2013 ad 2015 shows that implementation of the emission control plan reduced $NO_x$ and PM concentrations but resulted in a even worst $O_3$ pollutions. Such a phenomenon is quite interesting and should be highlighted in the abstract and conclusion. Does any satellite observation (such as OMI or TES on Aura) capture such a change of ozone in Eastern China?

**Response:** We have included a sentence in the abstract: "*Analyses of pollutants observations from 2013 to 2015 have shown that the concentrations of CO, $SO_2$, $NO_2$, and $PM_{2.5}$ from April to September in Eastern China have considerably decreased, but the $O_3$ concentrations have increased by 9.9%.*". We have also classified in the conclusion: "*Analyses of air pollutants observations in 66 cities from 2013 to 2015 have shown that, although implementation of the APPCAP has considerably decreased the CO, $SO_2$, $NO_2$, and $PM_{2.5}$ mass concentrations from April to September in Eastern China, the [$O_3$] have increased by 9.2% and the frequency of $O_3$ exceedance with hourly [$O_3$] exceeding 200 $\mu g\ m^{-3}$ has increased by about 25% in the afternoon.*"

Satellites have also observed the $O_3$ increasing trend in Eastern China from 2005 to 2014. We have clarified in Section 3.1: "*The ozone monitoring instrument (OMI) satellite observations have also shown that the annual $O_3$ concentration has increased by 1.6% per year over central and eastern China from 2005 to 2014 (Shan et al., 2016).*"

**2 Comment:** By turning off each emission source individually in the model, the authors tease out the role of each type of emission. One further question readers may have is what precursor species from each emission source are related to the ozone formation. It would be clearer if major VOC and NOx concentrations could be listed from each sector in the emission dataset used by the WRF-Chem model.

**Response:** We have added SI-Table 1 in the Supplementary Information (SI) to present the emission rate of major $O_3$ precursors from different sources and clarified in Section 3.3: "*SI-Table 1 further presents the emission rates of major $O_3$ precursors from different emissions sources in the model domain during the study episode. The industrial source dominates the VOCs and $NO_x$ emissions, playing a key role in the $O_3$ formation. The transportation source emits more NOx and active VOCs, such as olefins and aromatics, than the residential source, contributing considerably to the $O_3$ formation.*"

**3 Comment**: Authors mentioned the possible uncertainty from the simulated meteorological conditions to explain the model biases in reproducing ozone distribution. Would a nudging of surface wind and temperature be helpful to minimize the influence of meteorology?

**Response:** We have clarified in the conclusion: "*Meteorological conditions play a key role in the formation of air pollution, determining the formation, transformation, diffusion, transport, and removal of the air pollutants in the atmosphere (Bei et al., 2010, 2012). A nudging of wind and temperature fields using observations generally improves the simulation of meteorological fields, reducing the model biases in reproducing the $O_3$ temporal variation and spatial distribution. So future studies are needed to improve the*

*meteorological fields using the data assimilation, such as the four-dimension data assimilation (FDDA)."*

---

## Author Comment (AC3) · 21 Jan 2017

**Reply to Dr. Meiyun Lin (meiyun.lin@noaa.gov)**

We thank Dr. Meiyun Lin for the careful reading of the manuscript and helpful comments. We have revised the manuscript following the suggestion, as described below.

**Comment:** This is a nice paper reporting ozone pollution in China. I strongly encourage the authors to convert the unit for ozone mixing ratios from ug/m$^3$ to ppbv. The latter (ppbv) is a more common unit used in the literature for ozone. By converting the unit to ppbv, the readers can quickly get a sense of how severe ozone pollution is in China, as compared to the other parts of the world.

**Response:** We have clarified in Section 2.3: "*China MEP releases the pollutants observations using the mass concentration ($\mu g\ m^{-3}$ or $mg\ m^{-3}$) as the unit. Therefore, in order to keep consistent with the observations, the mass concentration is used in the manuscript, although the mixing ratio (such as ppbv) is a more common unit used in the literature for air pollutants.*" Generally, it is easy for the model to output species using the mass concentration or the mixing ratio. However, if the simultaneous measurement of the air density (or air temperature and pressure) at monitoring sites is not provided, the unit conversion from the mass concentration to the mixing ratio might be biased.